# Alveolar Ridge Preservation Using Three-Dimensional Root Replicas of Polycaprolactone: A Radiological and Histological Evaluation of a Case Report

**DOI:** 10.3390/reports8020092

**Published:** 2025-06-09

**Authors:** Pedro Christian Aravena, Mario E Flores, Larissa Córdova Turones, Francisca Pavicic, Pamela Ehrenfeld

**Affiliations:** 1School of Dentistry, Faculty of Medicine, Universidad Austral de Chile, Valdivia 5090000, Chile; 2Polymers Laboratory, Institute of Chemistry Sciences, Faculty of Sciences, Universidad Austral de Chile, Valdivia 5090000, Chile; mario.flores@uach.cl; 3Laboratory of Pathology, Faculty of Medicine, Universidad Austral de Chile, Valdivia 5090000, Chileingridehrenfeld@uach.cl (P.E.)

**Keywords:** 3D printing, case report, polycaprolactone, dental implant, scaffold, guided bone regeneration

## Abstract

**Background and Clinical Significance:** To describe the effectiveness of alveolar ridge preservation under the radiological and histological analysis of a customized resorbable scaffold three-dimensionally printed with polycaprolactone (PCL) reinforced with a coating of a copolymer of polycaprolactone-block-polyethylene glycol (PCL–PEG) by electrospray. **Case Presentation:** A 62-year-old male with vertical root fractures of teeth #14 and #15. From the cone beam CT (CBCT) image, the scaffold root replicas were designed with the shape of the roots and printed with PCL coated with PCL–PEG by electrospray. The scaffold was inserted into the alveolar bone and maintained with a tension-free flap closure. After six months, a CBCT of the surgical site and histological analysis of a bone sample at the dental implant installation site were performed. After 6 months, the wound in tooth #14 was closed, clinically proving no adverse reaction or complications. The histological analysis of the bone sample showed new bone formation with lamellar structure, Haversian canal structure, and osteocyte spaces. However, the scaffold in tooth #15 was exposed and not osseointegrated, and it was covered with membranous tissue. Histologically, the sample showed tissue compatible with lax connective tissue with mixed inflammatory infiltrate. In tooth #14, the dental implant presented an insertion torque >35 Ncm and was rehabilitated three months after its installation. **Conclusions:** Three-dimensional printed PCL scaffolds showed the ability to regenerate vital and functional bone with osseointegration capability for maxillary bone regeneration and oral rehabilitation based on dental implants. A case of inadequate scaffold osseointegration accompanied by lax connective tissue formation is shown.

## 1. Introduction and Clinical Significance

The most important risk factor for bone loss in the oral and maxillofacial area is tooth loss caused by the world’s most common oral chronic non-transmissible diseases: caries and severe periodontal diseases [1]. Tooth loss causes alveolar bone resorption, causing progressive alveolar bone atrophy, which reduces the chances of these patients’ prosthetic rehabilitation [2,3].

To this end, alveolar ridge preservation methods have been created using surgical techniques and bone-filling biomaterials such as xenografts, allografts, autografts, and/or synthetic materials used alone or in combination with a membrane to fill the dental socket immediately after tooth extraction [4]. However, volume preservation depends on the amount and resorption capacity of the biomaterial, as the presence of biomaterial remnants in the bone at 10 [5], 14 [6] and even 20 years [7] has been shown to generate a non-biological bone, with the presence of multinucleated giant cells around calcium crystals, with a scar-like acellular bone without lacunas typically seen between osteoclasts in reabsorbing bone surfaces [8,9].

At present, scaffolds to preserve and regenerate bone can be acquired through three-dimensional (3D) printing technology. Fused deposition modeling (FDM) with electrospinning is the most common technique to prepare fibrous materials possessing a non-woven structure, mimicking the structure of the extracellular matrix in bone tissues [10,11]. Nowadays, the most commonly used thermoplastic materials are medical-grade and biodegradable polymer polycaprolactone (PCL). PCL, an acronym for polylactide of poly(lactic acid), is commonly utilized as a printing material and is recognized as a biodegradable substance sanctioned by the Food and Drug Administration (FDA) for use as a bone-filling material in humans [12]. PCL is a highly soluble, synthetic biodegradable polyester, hydrophobic, semi-crystalline polymer with a low melting point, excellent biocompatibility, flexibility, and thermoplastic properties [13,14,15]. In addition, incorporating a bioactive copolymer phase such as propylene glycol (PGE) into polymer matrices creates bioactive hybrid composites that can greatly enhance biocompatibility, mechanical strength, and hydrophilicity [16,17], with improved osteogenic and angiogenic ability [18].

To date, preformed cone-shaped PCL has been used for socket preservation, proving to be a material that allows for bone regeneration. In 2015, Goh et al. [19] presented the efficacy of socket preservation using preformed PCL, demonstrating with histology new bone formation and subsequent installation with the osseointegration of a dental implant. This result contributes to the possibility of creating a customized scaffold with the shape of the post-extraction socket by the CAD-CAM process. This scaffold will facilitate the rapid development of new, densely mineralized mature bone that replicates the shape, size, and volume of the originally grafted matrix, while remaining economically feasible for patients.

This case report presents the effectiveness of a 3D-printed matrix designed with the anatomical shape of the dental socket and printed with PCL reinforced with a coating of a copolymer of polycaprolactone-block-polyethylene glycol (PCL–PEG) for socket preservation after tooth extraction and bone formation with subsequent implant installation and dental rehabilitation.

## 2. Case Presentation

### 2.1. Ethical Aspects

This case is part of a clinical trial established in accordance with the PROCESS writing guideline [20], and its protocol was reviewed and approved by the scientific ethics committee of the health service of Valdivia, Chile, belonging to the Chilean Ministry of Health (ORD: 541-2023), and with the clinicaltrials.gov registry number NCT06773923.

### 2.2. Case Report

A 62-year-old male patient consulted the dental clinic for a crown fracture of the right upper first and second premolar (#15 and #14) with a previous history of endodontic treatment (1 year ago) for irreversible pulpitis. He had no chronic medical illnesses, did not smoke, and had no allergies to medications or other environmental elements. On dental examination, he presented good oral hygiene and a tooth crown fracture, with the exposure of the canal treated with previous endodontics (Figure 1a).

A vertical fracture of the dental roots was detected on the CBCT scan (Figure 1b), prompting the decision to extract the tooth roots for subsequent bone regeneration and the installation of dental implants.

### 2.3. Preparation of 3D Scaffold

A 3D replica of the roots was created from radiographic images obtained from the patient by CBCT with a KaVo OP 3D Pro device (KaVo Dental GmbH, Biberach, Germany) before surgery. The scan had a field of view of 5 × 5 cm to 13 × 15 cm in diameter (scan parameters: 90 kV, 5 mA, 8.14 s exposure time, voxel size of 0.38 mm). The DICOM files were collected and imported into the MeshMixer software Version 3.5.0 (Autodesk, San Rafael, CA, USA) to model the shape of the root scaffold of teeth #14 and #15, which were then exported into a format suitable for 3D printing (Standard Triangulation Language, .stl) (Figure 1c). To produce a porous scaffold, the reconstructed volume was treated using cutting software (Ultimaker, Utrecht, The Netherlands). The gyroid filling pattern was selected because this pattern allows for a network of interconnected pores with a density of 70%, with a distance of 600 µm between the printed filaments, which represents the most suitable option within the capabilities and limitations of the software to approximate the natural porosity of bone [21]. Next, the digital files were converted into a printable format (.gcode) compatible with fused deposition modeling (FDM) printers. The scaffolds were created in a 1:1 ratio compared to the original roots using a Creality Ender-3 3D printer (Creality, Shenzhen, China), using PCL filament (1.75 mm diameter, Canada Filament Store), following specifications established in reported protocols [22], resulting in structures resembling dental roots (Figure 1e). The 3D-printed scaffolds were subsequently covered with an electrospray layer containing a block copolymer of PCL–PEG, the previously described protocol [22]. Finally, the printed matrices were sterilized in a lamellar flux camera, exposing scaffolds to UV light for 1 h.

### 2.4. Step 1: Surgery Tooth Extraction and Scaffold Placement

An oral surgeon with over 10 years of experience (P.C.A.) performed the root extraction surgery atraumatically according to the previously described surgical protocol [23]. Prior to surgery, the patient took a 1 g capsule of amoxicillin orally one hour before surgery and rinsed with a 0.12% chlorhexidine digluconate mouthwash (PerioAidTM 0.12%, Dentaid, Barcelona, Spain), and the skin and lips were cleaned with a 2% chlorhexidine solution, isolating the mouth exclusively using a sterile perforated surgical drape of 10 cm × 10 cm. Local infiltrative anesthetic was administered in the vestibular and palatal regions surrounding teeth #14 and #15 with two tubes of lidocaine HCl 2% and epinephrine 1:100,000 (Lignospan^®^ Septodont, Saint-Maur-des-Fossés, France). Then, a full-thickness flap was made with two vertical discharges one centimeter mesially and distally to the teeth using a #15C scalpel (Hu-Friedy Mfg. Co., Chicago, IL, USA) to extract the root debris using an atraumatic technique with a Hu-Friedy F29 straight elevator (Figure 1d). The post-exodontic sockets were cleaned with saline, and the scaffolds were carefully inserted into the socket with #17 straight forceps with controlled pressure until they were self-supporting (Figure 1f). Once the scaffolds were stable, the flap was closed and the wound stabilized with four single sutures with nylon 4.0 (EthilonTM, Johnson & Johnson, New Brunswick, NJ, USA) (Figure 1g). All patients were instructed to take ibuprofen 400 mg every 8 h for three days and to use 0.12% chlorhexidine mouthwash twice a day for one week (PerioAidTM 0.12%, Dentaid, Barcelona, Spain). The patient was monitored after 7 days during the next four weeks, and no dehiscence or wound infection was observed (Figure 1h).

### 2.5. Step 2: Bone Biopsy Surgery and Implant Installation

Six months after surgery, a control CBCT was performed with radiological characteristics similar to those at the beginning of the treatment and then the implant installation surgery, with anesthetic, surgical preparation, and a pharmacological protocol similar to the scaffold installation. In the control, in tooth #15, an exposure of the scaffold was observed (Figure 2a). Conversely, the area of tooth #14 presented a keratinized gingiva with no signs of infection or matrix exposure. A linear incision and a full-thickness mucoperiosteal flap were made with buccal displacement. In the area of tooth #14, there was abundant bleeding with a hard surface consistency similar to that of cortical bone (Figure 2b). At the site, a bone sample was taken using a trephine bur with an internal/external diameter of 2.0/3.0 mm, respectively, with an abundant irrigation of 0.9% saline (Figure 2c). An IntraOss^®^ CM Advanced implant, 3.5 m diameter and 11.5 mm long (IntraOss, SP, Brazil), was installed in the space according to the commercial installation protocol (Figure 2d). On the other hand, the matrix installed in tooth #15 was excised (Figure 2e), and the same implant was installed with osteotomy in the mesial of tooth #16. The space was closed with Creos Xenoprotec^®^ collagen membrane (Nobel Biocare, Zurich, Switzerland), and the flap was closed with a nylon 4.0 suture. Postoperative indications and the frequency of controls were similar to the first step of the previous surgery.

### 2.6. Radiological Study

To analyze the radiological presence of new bone formation, a CBCT was taken six months after scaffold grafting prior to implant installation surgery and four months after implant installation. The average value of Hounsfield units (Hu) at three points (apical, middle and coronal) of the grafted area was compared with that of the medullary bone located mesially to the mesiobuccal root of tooth #16 and distally to the root of tooth #16 using the “Density Value Measure [Hu]” tool available in the open-source implant planning software Blue Sky Plan v14.0 (Blue Sky Bio LLC; Grayslake, IL, USA) (Figure 3a).

### 2.7. Histological Study

The specimens were immersed in a 4% paraformaldehyde solution at room temperature for 48 h and then decalcified and sectioned into paraffin blocks. Twelve 5 μm thick serial sections were obtained from each specimen parallel to the long axis of the cylindrical core using an automatic microtome (Jung Supercut 2065, Leica Instruments GmbH, Heidelberg, Germany). The sections were mounted on slides and stained with Harris’s hematoxylin-eosin (H&E) and Masson trichrome staining. The histomorphometric analysis was performed by an expert in oral pathology in the Department of Oral Pathology at the Universidad de Chile, Chile, using an optical microscope (Optical microscope Olympus BX43TP. Olympus, Tokyo, Japan) at 10× and 40× magnification. Images were selected and transferred to a computer display through a digital camera attached to an optical microscope enabled for histomorphometric analysis.

### 2.8. Surgical Follow-Up

The surgery and scaffold installation were performed with no complications. Postoperative controls showed no wound dehiscence, pain, or inflammation. Healing was monitored at 7, 14, and 28 days, presenting complete wound closure (Figure 1h). At 6 months, exposure of the scaffold of tooth #15 was observed, with orange-colored edges, without bleeding or pain symptoms (Figure 2a). Given the clinical clarity of the lack of integration of the graft, the decision was made to remove the scaffold, noting dense fibrous tissue around it (Figure 2e). In contrast, in the sector of tooth #14, the scaffold was observed without clinical distinction, with abundant bleeding and firmness on palpation, with a consistency similar to that of cortical bone (Figure 2b).

### 2.9. Radiology and Histology Results

At six months, the CBCT of tooth #14 zone showed a radiographic image consistent with medullary bone, with an average scale of Hu: 321.5 ± 98.3 (Figure 3a), similar to the Hu unit of mesial tooth #16 (Hu: 371.9 ± 72.3) and distal tooth #13 (Hu: 369.5 ± 85.7).

In the histological sample of scaffold bone grafted in tooth #14, bone tissue was observed with the presence of a laminar structure with Haversian canals, longitudinal and concentric lamellae, and osteocyte spaces. No soft tissue or inflammatory cellular elements were observed (Figure 4a,b). On the other hand, the histological sample of the tissue extracted from the scaffold of tooth #15 presented a lax connective tissue with a mixed inflammatory infiltrate of lymphoplasmacytic predominance and the presence of small caliber blood vessels, and it was partially lined by a non-keratinized, pluristratified flat epithelium of hyperplastic appearance with atrophic areas (Figure 5a,b).

## 3. Discussion

The technique of socket preservation using graft fillings has historically been used to preserve the size and volume of cortical bone and gingiva, thereby avoiding the physiological resorption of already proven tissues [1,24]. Using autologous material, allograft, xenograft, alloplastic material, and even the clot itself has been effective [23,25,26]. However, the use of 3D-printed matrices of PCL customized to the shape, size, and volume of the alveolus has not been tested to date. In this report, we showed that the use of a PCL scaffold printed according to the anatomical shape of tooth roots and grafted into the socket allows for the preservation of alveolar tissue with a radiological and histological formation of new lamellar bone, thus enabling the installation of a dental implant and achieving osseointegration. At the same time, we present the first result of the lack of integration of a scaffold in alveolar bone and the histological formation of lax connective tissue with mixed inflammatory infiltrate around it.

Moreover, 3D-printed bone matrices are an option to contain bone filler material for bone regeneration in the jaws. To date, research in bone regeneration has shown that PCL can be used as a scaffold in tissue regeneration as it exhibits biocompatibility characteristics and the ability to be molded into porous structures that can support bone growth [27]. The scaffold designed and printed in replica form of the dental roots in our case was designed and printed with a porosity of 70% and a pore size of 600 µm, approximating the natural density porosity of bone [21]. These dimensions facilitate macrophage infiltration and promote the influx of additional cell growth factors essential for tissue colonization, migration, and vascularization in vivo [28]. Moreover, coating the scaffold with PGE by electrospray layer technique conferred wettability properties. It has been observed that the presence of co-polymer particles on the PCL surface reduces the water-in-air contact angle [17] and provides higher surface roughness—a key factor in protein adsorption regulation, capable of influencing cell fate and improving cell behavior, especially in terms of adhesion, proliferation, and differentiation [22,29]. As the new bone develops, the PCL–PEG scaffold degrades and is resorbed in distinct phases—the hydration phase (6 months), degradation and mass loss (6 to 12 months), resorption (post 12 months) and metabolization (post 18 months)—until it is completely replaced by regenerated tissue [30], demonstrating that the metabolic remains are completely excreted from the body [31] and are proven to be biologically safe materials for humans.

A scaffold of PCL has already been used in humans for the reconstruction of defects or tumors in long bones, in calvaria [32,33], lumbar vertebral body fusion [34], zygomatic bone reconstruction [35], maxillary bone and periodontal tissue repair [36] and lip and palate cleft closure [37]. Goh et al. [19] demonstrated new bone formation and achieved case rehabilitation with dental implants. On the other hand, a recent case report by Ivanovski et al. [38] demonstrated the ability of bone regeneration using a PCL matrix in tooth #2.1, showing the potential for the customized and personalized application of PCL in human bone segment regeneration.

The causes of the lack of integration of the scaffold of tooth #15 are diverse. The scaffold insertion surgery involved covering the surgical site by repositioning the flap that connects the attached gingiva of the buccal and palatal areas. Tension in the tissue was alleviated using horizontal incisions in the periosteum, facilitating a tension-free closure, which was confirmed by the complete closure of the wound in postoperative follow-ups (Figure 1h). However, after 6 months, exposure of the scaffold in the mouth was observed. Among the possible causes, the matrix exposure to the mouth allowed for the migration and formation of epithelial and subcutaneous cellular tissue at the bone–scaffold interface, and we histologically observed a fibrous connective tissue layer instead of bone [39]. This tissue reaction is similar to that demonstrated from the exposure of collagen membranes in guided bone regeneration (GBR) techniques, where a layer of fibrous connective tissue is observed between the recipient bone and the graft, rather than newly formed trabecular bones in contact with each other [40]. Another cause is the reaction of the mucosa to the implantation of the scaffold, generating epithelial hyperplasia, where the epithelium proliferates to reestablish the protective barrier; such responses are essential to maintain oral health and prevent infections [41]. On the other hand, the scaffold also has an effect on granulation tissue formation, as multinucleated giant cells induced by bone substitute types have been shown to produce conversion to the foreign body giant cell (FBGC) lineage, as they express only β-2 integrin; this is in contrast to osteoclasts outside the immediate implantation sites, which demonstrate only β-3 integrin expression [42]. Despite the loss of scaffold integration, the case was resolved by installing a dental implant in the area of the upper first molar and effectively rehabilitating the patient’s teeth (Figure 3b), maintaining the esthetics and function of the area, similar to that observed in previous reports [43] with implant installation in the socket 4 months after tooth extraction.

Limitations of this study are inherent to the case report design, such as the lack of a control group to compare the efficacy of the new PCL–PEG scaffold vs. other biomaterials on the market; the random bias of the results due to patient-specific factors (systemic health, healing ability, oral hygiene habits); possible measurement bias due to the fact that histological analysis may not reflect the entire alveolus or bone interface, limiting conclusions on overall bone regeneration; and the short follow-up time to corroborate the osseointegration of the dental implant, which does not allow for confirmation of the prediction of successful dental rehabilitation using the new scaffold. Despite these limitations, we consider it important to present this finding of using a 3D-printed dental root replica with PCL–PEG as a filling material for socket preservation and scaffold for bone formation.

## 4. Conclusions

Considering the limitations of this study design, we present a case report using socket preservation with PCL–PEG scaffolds designed as dental root replicas, showing their clinical, radiographic, and histological results in preserving bone volume, generating a vital bone, and enabling the osseointegration of a dental implant. At the same time, it is possible to demonstrate the tissue behavior without scaffold osseointegration by analyzing its possible biological causes. Future clinical trials are needed to determine the clinical and histological effectiveness of PCL–PEG in alveolar preservation and bone segment regeneration in the oral and maxillofacial field.

## Figures and Tables

**Figure 1 reports-08-00092-f001:**
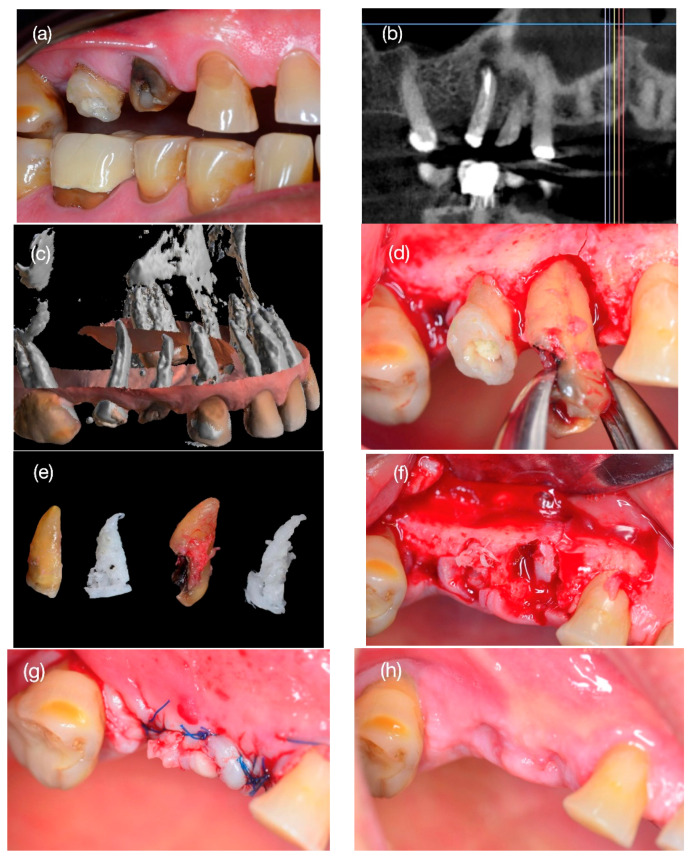
Condition of teeth #15 and #14 with vertical fracture (**a**) and presence of the root condition with osteolytic lesion in the apex of tooth #15 with poor prognosis for rehabilitation (**b**). With the CBCT exam, the three-dimensional reconstruction of the roots (**c**) and the creation of the scaffolds simulated the shape of the roots (**e**). In the tooth extraction (**d**), the scaffolds were installed in each socket (**f**), and the wound with a simple suture without flap tension was closed (**g**). After 28 days, the wound was closed without signs of infection, inflammation, or dehiscence (**h**).

**Figure 2 reports-08-00092-f002:**
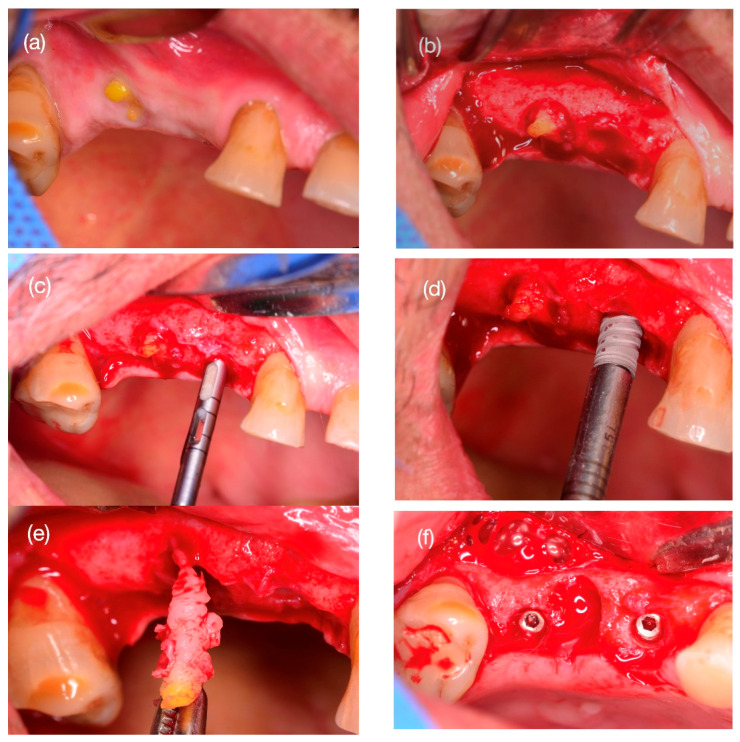
Second surgery. At 6 months, the case was checked, and exposure of the scaffold of tooth 15 and healthy and closed gingiva in the area of tooth 14 (**a**) was observed. At the opening of the full-thickness flap, integration of the scaffold of tooth 14 was noted with abundant bleeding and hard consistency of the surface similar to cortical bone. In tooth 15, the scaffold was not integrated (**b**). In tooth 14, the bone sample was taken with a trephine (**c**), and a dental implant of 3.5 mm × 11.5 mm (**d**) was installed. The scaffold of tooth #15 was excised (**e**), and a surgical bed was prepared in the distal part of the area, achieving an insertion torque of 35 Ncm (**f**) for both implants.

**Figure 3 reports-08-00092-f003:**
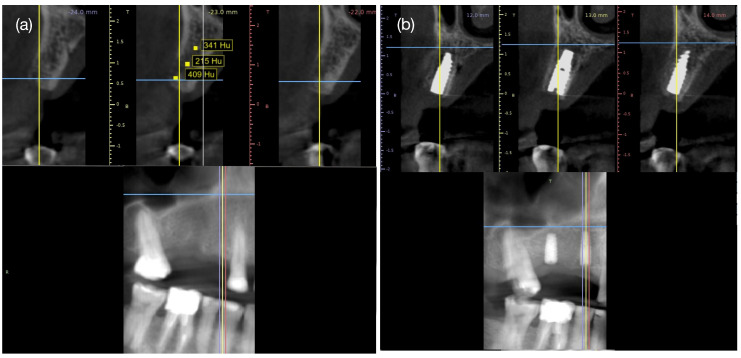
Radiographic control with CBCT six months after scaffold installation (**a**) and four months after dental implant installation (**b**). The colored lines on the radiographic image represent the tomographic slice every one millimeter. Radiological signs of medullary bone with Hounsfield unit (Hu) in the scaffold area (Hu: 321.5 ± 98.3) similar to medullary bone distally to tooth #13 (Hu: 369.5 ± 85.7) and mesially to tooth #16 (Hu: 371.9 ± 72.3). The dental implant shows signs of osseointegration and new bone formation covering the implant platform.

**Figure 4 reports-08-00092-f004:**
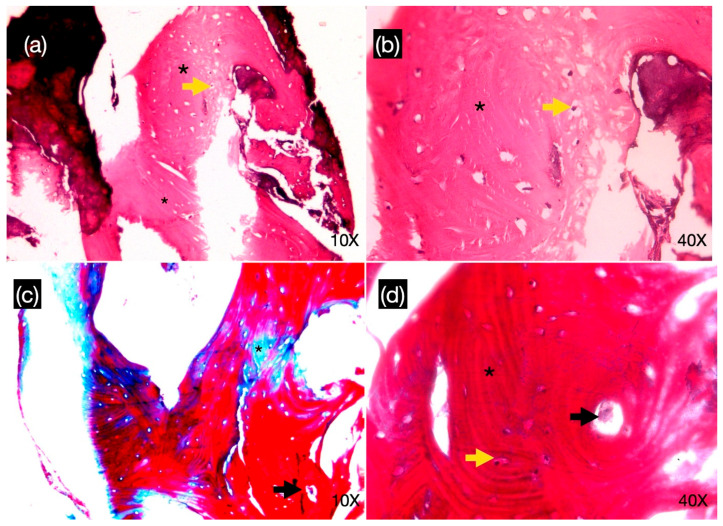
Histology with hematoxylin-eosin (**a**,**b**) and Masson Tricomic stain (**c**,**d**) from biopsy of tooth #14 area. Bone tissue was observed with the presence of lamellar structure (black asterisk) with the appearance of Haversian canals (black arrow), longitudinal and concentric lamellae, and osteocyte spaces (yellow arrow).

**Figure 5 reports-08-00092-f005:**
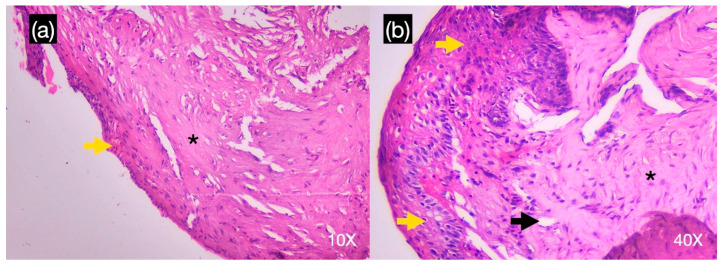
A histological sample with hematoxylin-eosin in 10× (**a**) and 40 × (**b**) of the tissue extracted from the scaffold of tooth #15 showed lax connective tissue with mixed inflammatory infiltrate of lymphoplasmacytic predominance (black asterisk), the presence of small caliber blood vessels (black arrow), and partially lined by nonkeratinized pluristratified flat epithelium of hyperplastic appearance with atrophic areas (yellow arrow).

## Data Availability

The original contributions presented in the study are included in the article, further inquiries can be directed to the corresponding author.

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
