# Peer review of "Alveolar Ridge Preservation Using Three-Dimensional Root Replicas of Polycaprolactone: A Radiological and Histological Evaluation of a Case Report"

_reports, 2025, doi:10.3390/reports8020092_

Round 1
Reviewer 1 Report
Comments and Suggestions for Authors
I reviewed the manuscript “Alveolar Ridge Preservation Using 3D Root Replicas of
Polycaprolactone. A Radiological and Histological Evaluation of Case Report.”.
Below I reported some doubts/suggestions for each manuscript’s sections both on
presentational and technical/communication aspects:
Introduction:
- Add an adequate reference at the end of the first sentence.
- The introduction should also include the current knowledge about this technique. Are
there other similar studies in the literature? Have other roots been printed in PCL for
alveolar ridge preservation? And in other materials? What is the background in the
literature on this technique, and what are the gaps?
Materials and Methods:
- Specify if the roots were printed in a 1:1 ratio compared to the natural roots.
Results:
- Line 197: It is not correct to speak of osseointegration as a biopsy and a microscopic
evaluation would be necessary. Since this is a clinical observation in this case, it is
necessary to rephrase the sentence. Please check this aspect in the entire manuscript.
- MAJOR CONCERN CONCERNING THE MANUSCRIPT: The scaffolds were printed with
the aim of preventing post-extraction bone loss. However, no measurements in mm of
the available bone volume before extraction and at the time of implant placement were
provided. Consequently, the report lacks the main information that would actually allow
for a discussion on whether the technique used is effective or not in preventing post-
extraction bone loss.
Conclusions:
- The conclusions should be presented separately from the discussion and rephrased.
The Authors indeed mention a clinical, radiographic, and histological success. However,
out of two scaffolds used, one was removed due to a lack of integration, while the other
was integrated. A positive result of 1 out of 2 cannot be generalized by speaking of
clinical and radiographic success (for which, moreover, the data that needs to be added
as previously mentioned is missing to be able to say so).
Minor considerations:
- Change in all the manuscript sections, the term “implant installation” to “implant
placement”.
- Remove the dots in the title.
- Bibliography is not formatting according to the journal’s guideline
Author Response
Comments1: - Add an adequate reference at the end of the first sentence.
Responses 1: We appreciate the reviewer's comment. We have added a new reference on the prevalence of caries and periodontal disease (ref: Kassebaum NJ, et al. . J Dent Res. 2017;96(4):380-38).
Comment 2: - The introduction should also include the current knowledge about this technique. Are there other similar studies in the literature? Have other roots been printed in PCL for alveolar ridge preservation? And in other materials? What is the background in the literature on this technique, and what are the gaps?
Response 2: If there are similar studies in in vitro and in-vivo studies (citations n°16-18). There are human reports of root printing (citation n°19) which are explained in detail in the discussion item of this article.
Comment 3: - Specify if the roots were printed in a 1:1 ratio compared to the natural roots.
Response 3: We appreciate the detail of this comment. We have added that the roots were printed in a 1:1 ratio.
Comment 4: - Line 197: It is not correct to speak of osseointegration as a biopsy and a microscopic evaluation would be necessary. Since this is a clinical observation in this case, it is necessary to rephrase the sentence. Please check this aspect in the entire manuscript.
Response 4: In the Methodology and Results section we present the bone biopsy with a microscopic evaluation obtained six months after the installation of the scaffold in the socket (Figure 4). It shows vital and functional bone tissue with the presence of lamellar structure (black asterisk) with the appearance of Haversian canals (black arrow), longitudinal and concentric lamellae, and osteocyte spaces (yellow arrow). This histologic specimen demonstrates that the scaffold achieved osseointegration at the surgical site. The biological justification is explained and detailed with the references n° 19, 26,28 and 30 in the discussion item.
Comment 5: - The scaffolds were printed with the aim of preventing post-extraction bone loss. However, no measurements in mm of the available bone volume before extraction and at the time of implant placement were provided. Consequently, the report lacks the main information that would actually allow for a discussion on whether the technique used is effective or not in preventing post extraction bone loss.
Response 5: We appreciate the reviewer's concern. In our opinion, the measurement of post extraction socket preservation is a topic that should be measured with volumetric and not linear instruments. This topic was presented in our publication (citation no. 22) for which it was necessary to scan the surgical site intraorally and measure the difference in volume before and after surgery. The aim of this study was different: to demonstrate that the use of a PCL as a material used for scaffold printed according to the anatomical shape of tooth roots and grafted into the socket allows the preservation of alveolar tissue with a radiological and histological formation of new lamellar bone, thus enabling the installation of a dental implant and achieving osseointegration.
Comment 6: - The conclusions should be presented separately from the discussion and rephrased.
Response 6: We have separated the conclusion from the discussion item.
Comment 7: -The Authors indeed mention a clinical, radiographic, and histological success. However, out of two scaffolds used, one was removed due to a lack of integration, while the other was integrated. A positive result of 1 out of 2 cannot be generalized by speaking of clinical and radiographic success (for which, moreover, the data that needs to be added as previously mentioned is missing to be able to say so).
Response 7: We appreciate the reviewer's detail. As investigators we wanted to present the lack of integration of tooth #15 scaffold to demonstrate the possibility of lack of osseointegration due to local and surgical factors. We also discussed the biological causes (citations: 39 to 42) and how to prevent its exposure following its surgical installation. All this is detailed in the discussion item of the manuscript.
Comment 8: Minor considerations:
- Change in all the manuscript sections, the term “implant installation” to “implant placement”.
- Remove the dots in the title.
- Bibliography is not formatting according to the journal’s guideline
Response 8: We have changed the reviewer's minor considerations.
Reviewer 2 Report
Comments and Suggestions for Authors
Dear Authors,
The present study is a case report reporting both radiological and histological results of a tridimensional root replicas in the alveolar ridge preservation.
The topic is innovative and of interest for potential readers, however I would raise the following:
- In the surgical procedure there is not mention of an antibiotic procedure. Was it done? Please specify in both cases.
- There is no limit section in the discussion, while it would be important for potential readers to have a clear idea about the limitations of a case report.
- To enrich the discussion I would consider this notable scientific article (PMID: 27722222)
- The conclusion session is not divided from the discussion and this may generate confusion for readers
- I would avoid strong terms as "demonstrate" being the present article a case report, and I would better underline the need of randomized controlled clinical trials on the present topic
Author Response
Comment 1: In the surgical procedure there is not mention of an antibiotic procedure. Was it done? Please specify in both cases.
Response 1: We appreciate the reviewer's question. We add that patient take a dose 1 g amoxicillin orally one hour before surgery.
Comment 2: There is no limit section in the discussion, while it would be important for potential readers to have a clear idea about the limitations of a case report.
Response 2: We appreciate this observation. We have added a paragraph on the limitations of the study at the end of the discussion section.
Comment 3: To enrich the discussion I would consider this notable scientific article (PMID: 27722222)
Response 3: We have added the interesting result presented by Felice P. et al (2016) at the end of the discussion item.
Comment 4: The conclusion session is not divided from the discussion and this may generate confusion for readers
Response 4: We have separated a following section for the item conclusion.
Comment 5: I would avoid strong terms as "demonstrate" being the present article a case report, and I would better underline the need of randomized controlled clinical trials on the present topic
Response 5: We agree with the reviewer's opinion. We have removed the word “demonstrate” in our main text.
Reviewer 3 Report
Comments and Suggestions for Authors
Thanks for an interesting case report with two different outcomes in the adjacent teeth area. It could have been more convincing if at least three cases presented. The introduction, methodology, results are fine. The discussion lacks the limitation of the study (case report). You can also discuss the cost factor, availability of materials and comparison of PCL-PEG effectiveness with other graft materials/ scaffold materials.
Author Response
Comment 1: Thanks for an interesting case report with two different outcomes in the adjacent teeth area. It could have been more convincing if at least three cases presented. The introduction, methodology, results are fine. The discussion lacks the limitation of the study (case report). You can also discuss the cost factor, availability of materials and comparison of PCL-PEG effectiveness with other graft materials/ scaffold materials.
Response 1: We appreciate the reviewer's comment. We have added a full paragraph on study limitations and some topics that you can read before the Conclusion item.
Reviewer 4 Report
Comments and Suggestions for Authors
This case report of socket preservation using the PCL-PEG scaffolds designed as the dental root replicas demonstrated its clinical, radiographic, and histological success in preserving bone volume, generating viable bone, and achieving osseointegration of dental implants. These results will aid in the continued development of future clinical trials. I have no objection to this manuscript.
Author Response
Comment 1: This case report of socket preservation using the PCL-PEG scaffolds designed as the dental root replicas demonstrated its clinical, radiographic, and histological success in preserving bone volume, generating viable bone, and achieving osseointegration of dental implants. These results will aid in the continued development of future clinical trials. I have no objection to this manuscript.
Response 1: We appreciate the reviewer's comments. We look forward to continuing to contribute future results.
Round 2
Reviewer 1 Report
Comments and Suggestions for Authors
- I prefer that Authors improve the first sentence as follows: "The most important risk factor for bone loss in the oral and maxillofacial area is tooth loss caused by the world's most common oral chronic non-transmissible diseases: caries and severe periodontal diseases" because caries and perdiontal diseases are not the most common in the world compared to the cardiovascular diseases for example.
- The authors replied that they have already detailed the results of previous studies using this technique in the introduction. However, I keep seeing only this sentence about the in vivo studies "To date, preformed cone-shaped PCL has been used for socket preservation, proving to be a material that allows bone regeneration.", in my opinion, this remains insufficient to provide an adequate background for the reader. The main results of previous studies should be truly presented in detail in the introduction.
- It was clear that a biopsy had been performed. However, in some statements, such as in line 199, osseointegration is mentioned during the clinical examination of the oral cavity! It is in these cases that the term osseointegration has been used improperly and should be corrected. When the authors refer to osseointegration following the biopsy, no correction is obviously necessary. However, I repeat, if this observation is made before the biopsy, the term is improper and must be corrected.
Author Response
Comment 1: I prefer that Authors improve the first sentence as follows: "The most important risk factor for bone loss in the oral and maxillofacial area is tooth loss caused by the world's most common oral chronic non-transmissible diseases: caries and severe periodontal diseases" because caries and perdiontal diseases are not the most common in the world compared to the cardiovascular diseases for example.
Response 1: We have added the word “oral” in the Introduction section of main text.
Comment 2: The authors replied that they have already detailed the results of previous studies using this technique in the introduction. However, I keep seeing only this sentence about the in vivo studies "To date, preformed cone-shaped PCL has been used for socket preservation, proving to be a material that allows bone regeneration.", in my opinion, this remains insufficient to provide an adequate background for the reader. The main results of previous studies should be truly presented in detail in the introduction.
Response 2: We appreciate the reviewer's detail. We add the sentence: “In 2015, Goh et al.[19] presented the efficacy of socket preservation using preformed PCL, demonstrating with histology the formation of new bone and the subsequent installation with osseointegration of a dental implant.” With that, we substantiate the possibility of creating a scaffold designed according to the anatomical shape of the dental root for socket preservation with PCL-PEG.
Comment 3: It was clear that a biopsy had been performed. However, in some statements, such as in line 199, osseointegration is mentioned during the clinical examination of the oral cavity! It is in these cases that the term osseointegration has been used improperly and should be corrected. When the authors refer to osseointegration following the biopsy, no correction is obviously necessary. However, I repeat, if this observation is made before the biopsy, the term is improper and must be corrected.
Response 3: We have removed the word “osseointegration” from the original line n°199. We have only stated the clinical observation of the scaffold after flap opening of the surgical site (Figure 2b).
Reviewer 2 Report
Comments and Suggestions for Authors
The Authors addressed the comments and observations arisen, improving the quality of the manuscript.
Author Response
Comment 1: The Authors addressed the comments and observations arisen, improving the quality of the manuscript.
Response 1: We appreciate your commentaries and consideration for publish our case reports. Thank you.
Round 3
Reviewer 1 Report
Comments and Suggestions for Authors
I thank the Authors for the revisions.